# Collective excitations of a bound-in-the-continuum condensate

Anna Grudinina ®[1], Maria Efthymiou-Tsironi ®[2,3], Vincenzo Ardizzone[2,3], Fabrizio Riminucci ®[4], Milena De Giorgi ®[3], Dimitris Trypogeorgos ®[3], Kirk Baldwin[5], Loren Pfeiffer[5], Dario Ballarini ®[3], Daniele Sanvitto ®[3] ✉ & Nina Voronova ®[1] ✉

Spectra of low-lying elementary excitations are critical to characterize properties of bosonic quantum fluids. Usually these spectra are difficult to observe, due to low occupation of non-condensate states compared to the ground state. Recently, low-threshold Bose-Einstein condensation was realised in a symmetry-protected bound state in the continuum, at a saddle point, thanks to coupling of this electromagnetic resonance to semiconductor excitons. While it has opened the door to long-living polariton condensates, their intrinsic collective properties are still unexplored. Here we unveil the peculiar features of the Bogoliubov spectrum of excitations in this system. Thanks to the dark nature of the bound-in-the-continuum state, collective excitations lying directly above the condensate become observable in enhanced detail. We reveal interesting aspects, such as energy-flat parts of the dispersion characterized by two parallel stripes in photoluminescence pattern, pronounced linearization at non-zero momenta in one of the directions, and a strongly anisotropic velocity of sound.

Bound states in the continuum (BICs) have been originally proposed as a mathematical feature of the Schrödinger equation in presence of specially prepared potentials[1–3]. Similar states have been later found in a wide range of systems like graphene, topological insulators, atomic superlattices, dielectric photonic crystals, metasurfaces and patterned optical waveguides[4]. In the latter cases, coupling in both the real and the imaginary parts of the two counter-propagating electromagnetic modes leads to the opening of an energy gap, with one of the new eigenmodes being bright and the other exhibiting vanishing radiative losses despite the non-Hermiticity of the underlying Hamiltonian[5,6]. Symmetry-protected optical BICs occuring from two interfering resonances enable light confinement and are routinely used in grating surface-emitting and distributed feedback lasers[7–9], where emission from the BIC mode manifests itself in a specific two-lobe far field pattern. Interestingly, these quasi-infinite-lifetime photonic states can couple to matter excitations, such as surface plasmons[6] or excitons in monolayer transition-metal dichalcogenides[10,11] and quantum wells[12] and form polaritons with interesting features, such as greatly enhanced tunable lifetimes and strong nonlinearities.

Polaritons are hybrid half-light half-matter quasi-particles in semiconductor heterostructures, typically embedded in a Fabry-Pérot microcavity to enhance the coupling of electronic excitations (e.g. excitons) with photons and to reduce the losses[13]. Such low-dimensional bosonic systems have been shown to provide conditions for observation of collective behaviors like Bose-Einstein condensation[14–16] and superfluidity[17–19], at the same time allowing for a direct measurement of their momentum and spatial distributions via the emitted light. The onset of Bose-Einstein condensation is

[1]National Research Nuclear University MEPhI (Moscow Engineering Physics Institute), 115409 Moscow, Russia. [2]Dipartimento di Matematica e Fisica "Ennio De Giorgi", Università del Salento, Strada Provinciale Lecce-Monteroni, Campus Ecotekne, Lecce 73100, Italy. [3]CNR Nanotec, Institute of Nanotechnology, via Monteroni, 73100 Lecce, Italy. [4]Molecular Foundry, Lawrence Berkeley National Laboratory, One Cyclotron Road, Berkeley, CA 94720, USA. [5]PRISM, Princeton Institute for the Science and Technology of Materials, Princeton University, Princeton, NJ 08540, USA. ✉e-mail: daniele.sanvitto@nanotec.cnr.it; neenoune@gmail.com

accompanied with a narrowed enhanced emission from the ground state at $k = 0$ of the polariton energy dispersion, which consequently blueshifts due to particle-particle interactions. At the same time, the momentum-energy dispersion on top of the condensate becomes linearized in accordance with the textbook Bogoliubov prediction[20], therefore giving origin to superfluidity as follows from the Landau criterion. Multiple experiments with cavity exciton-polaritons have provided evidence of such linearization[21–25] despite the driven-dissipative nature of the system suggesting the diffusive character of the Bogoliubov dispersion[26,27] with a zero real part in the region of small momenta. Precise observation of the shape of the excitations spectrum due to the thermal and quantum depletion of the macro-scopically occupied ground state is also possible, but requires sub-stantial momentum-space filtering covering a much brighter signal from the condensate[28,29] or using refined interferometric techniques[25].

In this work, we study the elementary excitation spectrum of a polariton Bose condensate arising from a BIC state in a planar nanos-tructured waveguide. It has been recently reported[12] that due to the band folding effect and the consequent coupling with the quantum well exciton mode, the condensate which is formed on the lowest polariton branch appears, counter-intuitively, at a saddle point of the dispersion rather than in a global energy minimum. Since the particles accumulate in the quasi-BIC state with a very long lifetime, much longer than the carriers relaxation time, the polariton condensate in such a state appears to be a paradigmatic system for studying the excitation spectrum of a condensate in thermal equilibrium[30]. In such state the condensate cannot directly radiate and, as we demonstrate in the following, measurements of the shape of the spectrum of ele-mentary excitations are possible in great detail and even without any momentum filtering. We derive theoretically the finite-temperature Bogoliubov dispersion for polaritons accumulating in the BIC and show that a local energy minimum starts to appear around the saddle

point $k = 0$ in the spectrum of excitations which, possibly, could help in the formation of a long-living Bose-Einstein condensate, with the real part of the spectrum being non-zero despite the negative effective mass in one of the directions and losses present in the system. We experimentally observe various and very distinct characteristics of collective excitations dispersion strongly dependent on the direction in $k$-space. The detailed knowledge of the Bogoliubov spectrum allows to investigate the angular profile of the sound velocity in such highly anisotropic system. We believe that this study could open a way of controlling the condensate properties by engineering its excitation spectrum.

## Results

### Saddle-point polariton condensate

The sample studied here, sketched in Fig. 1a, is a GaAs/AlGaAs waveguide hosting 12 20-nm-thick GaAs quantum wells embedded in a $Al_{0.4}Ga_{0.6}As$ core. A set of linear diffraction gratings is etched on the surface of the waveguide to allow for the observation, within the lightcone, of the propagating guided modes. By changing the grating period $a$ and the filling factor it is possible to tune the energy of the BIC (i.e. its excitonic fraction) and the width of the energy gap separating the lower-polariton BIC, with ultra-long lifetimes, and the leaky polaritonic modes. The cou-pling of the counter-propagating TE-modes of electromagnetic waves to the exciton results in the four branches of polariton dispersion, the analytical derivation of which are provided in the Supplementary Information (SI). Of interest, in this work, are the two anisotropic lowest branches which are shown in Fig. 1b. It is clearly seen that the shape of the upper of the two lower-polariton modes (ULP) resembles the regular lower cavity exci-ton-polariton, albeit with differing effective masses along and perpendicular to the grating, whereas the lower (LLP) mode is

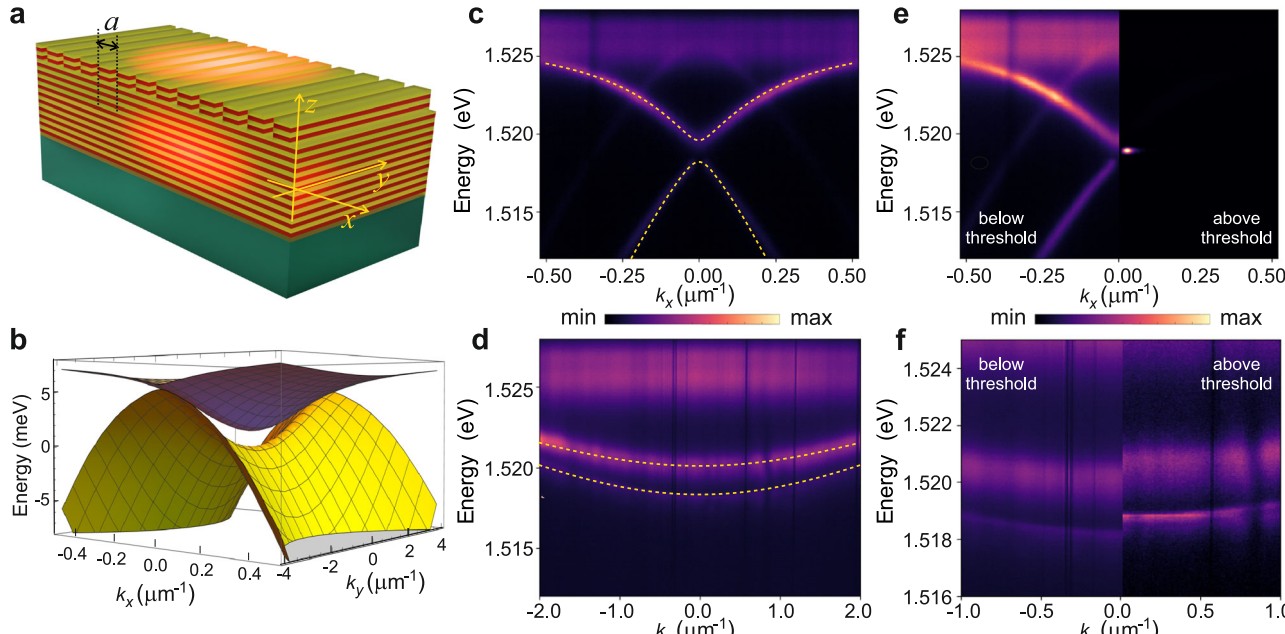

**Fig. 1 | Polariton dispersions around the BIC state. a** Sketch of the etched waveguide structure. The waveguide is made of 12 GaAs/$Al_{0.4}Ga_{0.6}As$ (20-nm-thick each) layer pairs on a substrate, with the top 90 nm textured with a periodic 1D grating with a period $a = 243$ nm. The grating sizes are 50 μm along $y$ and 300 μm along $x$-direction. **b** Three-dimensional dispersions of the two lower exciton-polariton modes (see SI), with the energy reference taken at the saddle point of the lowest mode, calculated for $\Delta_0 = -0.6$ meV, $\hbar\Omega_R = 15$ meV, $U = -1.3$ meV, $\hbar\gamma = 0.51$ meV, $n_g = 4.3$, $m_X = 0.22\, m_0$. **c, d** Cross-cuts of the energy-momentum experimental

dispersion along $k_y = 0$ (**c**) and $k_x = 0.025\, μm^{-1}$ (**d**) showing two orders of magnitude difference in the lower polariton effective mass along $x$ and $y$ directions, $1/m_{LP}^{x(y)}$ (see text). The yellow dashed lines show theoretical fits of the two lower polariton branches with the parameters as in **b**. **e, f** Comparison of experimental far field emission below (on the left-hand side) and above condensation threshold (on the right-hand side) showing energy versus $k_x$ and $k_y$ respectively; panel **f** is a cut along $k_x = 0.02\, μm^{-1}$, as for panel **d**. The colorscales for intensity are linear, given in arb. units and normalized separately for panels **c, d** and **e, f**.

distinct from the standard dispersion showing different behaviors along $k_x$- and $k_y$-directions. In particular, we note the appearance of a negative effective mass in the $x$-direction, with an absolute value much lower than the positive mass in the $y$-direction. It is also important to note that, when expanded at $k \to 0$ in Taylor series up to the second order, the LLP dispersion appears to have an imaginary contribution in the $x$-direction (for more details, see the SI). This imaginary term provides the $k_x$-dependent loss rate of polaritons on the lowest branch, with the zero radiative loss at $k = 0$, as was experimentally verified in ref. 12.

The polariton dispersions along $k_x$ and $k_y$ can be directly measured by energy and momentum resolved photoluminescence (PL). Figure 1c and d show two different cuts (along $k_x$ and $k_y$ respectively) of the PL of the single-particle lower polariton dispersions. While along $k_x$ (Fig. 1c) the ULP and LLP have the masses of opposite sign, this is not the case along $k_y$ where the mass is positive for both (Fig. 1d). The dispersion in Fig. 1d has been taken at a slightly finite wavevector in the $x$-direction ($k_x \sim 0.02\,\mu m^{-1}$) to avoid the completely dark stripe of the BIC state at $k_x = 0$. These experimental dispersions are in very good agreement with the analytical model which is shown by the overlaid yellow dashed lines in both panels. We note that the range of horizontal axis along $k_y$ is four times larger than that of $k_x$ and that the effective masses along the two directions differ by about two orders of magnitude, being in quantitative agreement with the theory developed in the SI. Both images in Fig. 1c and d show the PL emission below threshold, when the upper of the two lower-polariton branches is emitting more light than the lowest one, as it is much more lossy around $k = 0$ (see SI), while their populations are comparable. When increasing the excitation power, polaritons condense into the BIC at the saddle point of the LLP branch[12]. Figure 1e (right-hand side) shows the cross-cut of the dispersion along $k_x$ with the time-integrated PL measured slightly above the condensation threshold. We only show one of the two bright spots forming to either side of the saddle-point condensate, blueshifted compared to the below-threshold image shown on the left-hand side of the panel. Figure 1f (right-hand side) shows the corresponding dispersion along $k_y$ for $k_x \sim 0.02\,\mu m^{-1}$, i.e. close to the inner edge of the bright spot of Fig. 1e above threshold. The latter has an interesting form, with an energy-flattened emission roughly extending from $-0.5\,\mu m^{-1}$ (not visible) to $+0.5\,\mu m^{-1}$ continuing into linear tails going up in energy. It is important to note that, as we will show in the following, the flat region in this case is not related to the diffusive character of the condensate. On the contrary, it is a consequence of both the very long polariton lifetime and the saddle shape of the dispersion.

## Hartree−Fock−Bogoliubov theory for elementary excitations

To explain these features observed in Fig. 1e, f, we derive the Bogoliubov spectrum of excitations on top of the condensate forming in the BIC state, i.e. in the vicinity of the $k = 0$ saddle point of the LLP branch for low finite temperatures in the presence of the (dark) exciton reservoir[30]. The Hamiltonian of lower polaritons is expressed in the second quantization as

$$
\begin{aligned}
\hat{H} = &\int d\mathbf{r}\,\hat{P}_-^\dagger(\mathbf{r})[\varepsilon_-(\hat{\mathbf{p}}) - \mu_-]\hat{P}_-(\mathbf{r}) \\
&+ \int d\mathbf{r}\,\hat{P}_+^\dagger(\mathbf{r})[\varepsilon_+(\hat{\mathbf{p}}) - \mu_+]\hat{P}_+(\mathbf{r}) \\
&+ \frac{1}{2}\int d\mathbf{r}d\mathbf{r}'\,\hat{Q}^\dagger(\mathbf{r})\hat{Q}^\dagger(\mathbf{r}')U(\mathbf{r} - \mathbf{r}')\hat{Q}(\mathbf{r}')\hat{Q}(\mathbf{r}) \\
&+ \int d\mathbf{r}d\mathbf{r}'\,\hat{\tilde{Q}}^\dagger(\mathbf{r})\hat{Q}^\dagger(\mathbf{r}')U(\mathbf{r} - \mathbf{r}')\hat{Q}(\mathbf{r}')\hat{\tilde{Q}}(\mathbf{r}),
\end{aligned}
\tag{1}
$$

where $\hat{P}_\pm(\mathbf{r})$ are the ULP and LLP field operators, respectively, $\hat{\mathbf{p}} = -i\hbar\nabla, \varepsilon_\pm(\mathbf{p}) = E_\pm^{LP}(\mathbf{p}) - \text{Re}\,E^{LP}(0)$ are the corresponding bare particle dispersions (the shape of $E_\pm^{LP}(\mathbf{p})$ is given in the SI) counted from the saddle point of the LLP branch, $\mu_\pm$ is the chemical potential, $\hat{Q}(\mathbf{r}) = \int d\mathbf{r}'[X(\mathbf{r}' - \mathbf{r})\hat{P}_-(\mathbf{r}') + C(\mathbf{r}' - \mathbf{r})\hat{P}_+(\mathbf{r}')]$ is the exciton field operator, and $\hat{\tilde{Q}}(\mathbf{r})$ describes the field of background excitons that do not directly convert into polaritons (e.g. dark excitons). In the above, $X(\mathbf{r}' - \mathbf{r}) = (1/S)\sum_\mathbf{p} X_\mathbf{p} \exp\{i\mathbf{p} \cdot (\mathbf{r}' - \mathbf{r})/\hbar\}, C(\mathbf{r}' - \mathbf{r}) = (1/S)\sum_\mathbf{p} C_\mathbf{p} \exp\{i\mathbf{p} \cdot (\mathbf{r}' - \mathbf{r})/\hbar\}, X_\mathbf{p}$ and $C_\mathbf{p}$ are the exciton and photon Hopfield coefficients, $S$ is the quantization area. For simplicity, the exciton-exciton pair interaction potential is assumed to be contact: $U(\mathbf{r}) = g\delta(\mathbf{r})$. The Hamiltonian (1) allows to write the Heisenberg equations for the evolution of both lower polariton fields $\hat{P}_\pm(\mathbf{r},t)$.

To obtain the excitation spectrum on top of the Bose condensate of BIC polaritons, we will assume that above threshold the LLP is macroscopically populated, which is supported by the experimental observation (see Fig. 1e). Since the polaritons occupying the ULP branch do not convert into LLP polaritons because of the different symmetry of the underlying photon and exciton modes, we exclude the ULP operators from consideration when describing the macroscopically occupied saddle point. In this case, one can separate the condensate in the regular Bogoliubov fashion and, by averaging the Heisenberg equation for $\hat{P}_-$ in the Hartree-Fock mean-field approximation (see "Methods" section), obtain the expression for the chemical potential of the LLP polaritons:

$$
\mu_- = g|X_0|^2\left(n_0|X_0|^2 + 2n_Q' + \tilde{n}\right),
\tag{2}
$$

where $n_0$ is the condensate density, $n_Q' \equiv \langle\hat{Q}^\dagger(\mathbf{r})\hat{Q}'(\mathbf{r})\rangle$ is the non-condensate exciton density due to finite temperature, and $\tilde{n} \equiv \langle\hat{\tilde{Q}}^\dagger(\mathbf{r})\hat{\tilde{Q}}(\mathbf{r})\rangle$ is the background reservoir density. We note that $\mu_-$ contains the thermal contribution $n_Q'$ and the dark reservoir contribution $\tilde{n}$ compared to the regularly used expression for the lower-polariton chemical potential $\mu_{LP} = gn_0|X_0|^4$. Importantly, Eq. (2) corresponds to the experimentally-observed condensate blueshift, which will be used together with the excitation spectrum below to define $n_0$ and $\tilde{n}$ at a given pump power (note that at low temperatures considered here, $n_Q' \ll n_0$). Diagonalizing the Hamiltonian (1) within the Hartree-Fock-Bogoliubov framework, one obtains the excitation spectrum:

$$
E_\mathbf{p} = \sqrt{\mathcal{E}(\mathcal{E} + 2gn_0|X_0|^2|X_\mathbf{p}|^2)} \quad \text{with} \quad \mathcal{E} = \varepsilon_-(\mathbf{p}) + g(|X_\mathbf{p}|^2 - |X_0|^2)(n_0|X_0|^2 + 2n_Q' + \tilde{n}).
\tag{3}
$$

It needs to be underlined that while Eq. (3) has the form looking similar to the textbook Bogoliubov prediction, it is, in fact, substantially different. Besides noticing that $\mathcal{E}$ contains the reservoir contributions $n_Q', \tilde{n}$ and the renormalization due to the dependence of the Hopfield coefficient $X_\mathbf{p}$ on momentum, the bare polariton dispersion $\varepsilon_-(\mathbf{p})$ itself is very unusual: in the $k_x$-direction it features, at the same time, the negative effective mass and momentum-dependent photon losses (see Supplementary Fig. 1 and Eq. (8) in the SI). Due to the latter, despite the negatively-defined $\text{Re}\,E_\mathbf{p}^2$ in the vicinity of $k_x = 0$, the real part of Eq. (3) is nowhere zero except the special points of the dispersion where $\text{Im}\,E_\mathbf{p}^2$ changes sign. The excitation spectrum does not acquire a diffusive character predicted for systems with dissipation[26,27]. In fact, here the non-zero imaginary part of $\varepsilon_-(\mathbf{p})$ ensures the existence of low-lying states just above the condensate with $\text{Re}\,E_\mathbf{p} > 0$. The analysis of the imaginary part of the spectrum is given below.

Figure 2a shows the real part of the Bogoliubov dispersion of excitations (Eq. 3) on top of the condensate formed in the saddle point of the LLP branch. The linearization at small momenta in the $y$-direction is clearly seen. Figure 2b, c show the cuts of $\text{Re}\,E_\mathbf{p}$ in Fig. 2a

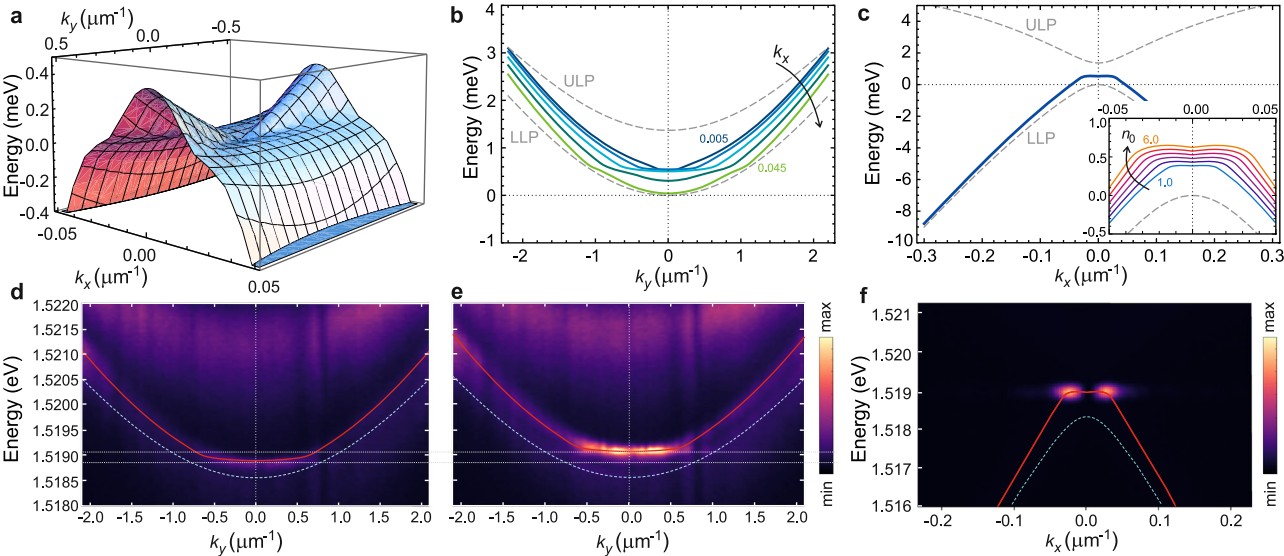

**Fig. 2 | Dispersion of elementary excitations of the saddle-point polariton condensate. a** The Bogoliubov dispersion of elementary excitations (real part of (3)) with added blueshift of the saddle point. **b** Cross-cut of **a** at different $k_x$ from 0.005 to 0.045 μm$^{-1}$ with the step 0.01 μm$^{-1}$, showing the flattening of the dispersion along the $k_y$–direction close to the BIC state at $k_x = 0$, calculated for condensate density $n_0 = 4 \times 10^{10}$ cm$^{-2}$, reservoir density $\bar{n} = 3 \times 10^{10}$ cm$^{-2}$. The gray dashed lines show the bare single-particle dispersions $\varepsilon_\pm(k_x = 0, k_y)$. **c** Cross-cut of **a** at $k_y = 0$, showing the real part of the Bogoliubov excitations spectrum $E_{\mathbf{p}}$ given by (3). The gray dashed lines show the bare single-particle dispersions $\varepsilon_\pm(k_x, k_y = 0)$. Inset shows a close-up of the local minimum of the excitations spectrum, for varying $n_0$ from 1 to $6 \times 10^{10}$ cm$^{-2}$. Parameters as on Fig. 1, $g = 2.5$ μeVμm$^2$, $T = 10$ K. **d, e** Experimental time-resolved photoluminescence images above threshold displaying the energy-momentum dispersion of excitations vs. $k_y$; the excitation spectrum is evident in both the flat part and the linearized tails. **f** Experimental PL image of the dispersion of excitations vs. $k_x$. In **d**–**f** the red lines show theoretical fitting of the Bogoliubov dispersion according to the real part of (3), with $n_0 = 4 \times 10^{10}$ cm$^{-2}$, $\bar{n} = 3 \times 10^{10}$ cm$^{-2}$ (**d**) and $n_0 = 5 \times 10^{10}$ cm$^{-2}$, $\bar{n} = 4 \times 10^{10}$ cm$^{-2}$ (**e, f**). The light-blue dashed lines show the single-particle LLP dispersion. The gray dotted lines in **d, e** mark the blueshift at different powers. The colorscales are in arb. units and normalized separately for **d, e** and **f**.

along different near-zero $k_x$ (b) and $k_y = 0$ (c), displaying the flattening of the dispersion in the $k_y$–direction. As can be anticipated from (3), the extent of the energy-flat part along $k_y$ depends on the chosen near-zero $k_x$, while the slope of the linear tails of the excitation spectrum at larger $k_y$ (for each $k_x$) is unambiguously defined by the polariton condensate density $n_0$. Moreover, the zoom-in on the small momenta region displays a local minimum in the $k_x$–direction (see the inset of Fig. 2c). At the values of $k_x$ outside this local minimum, the Bogoliubov spectrum recovers the negative slopes characteristic to the single-particle LLP dispersion.

### Excitations luminescence dynamics in the pulsed regime

The experimental PL images of collective excitations obtained in the regime of pulsed excitation are shown in Fig. 2d–f. While the features very close to $k_x \sim 0$ are below the experimental resolution and correspond to the dark region due to the long BIC lifetime, the dispersion along $k_y$ shows a striking correspondence to the theory derived above. To perform a better comparison, we have measured the temporal dynamics of the dispersion of excitations in PL along $k_y$. This allows us to rule out any temporal smearing of the dispersion and, at the same time, track the change of the dispersion for different condensate and reservoir densities. Figure 2d-e display two different snapshots of the emission along $k_y$ for $k_x \sim 0.02$ μm$^{-1}$. These snapshots corresponding to different times clearly exhibit an energy-flat region with less-pronounced linear tails at higher values of $k_y$. Thanks to the intrinsic dark nature of the condensate from a BIC state, it is possible to directly measure these features without any filtering in the momentum space. The red solid lines in Fig. 2d, e show the analytical cross-cuts of the dispersion (Eq. 3) along $k_x \sim 0.02$ μm$^{-1}$, for two values of densities calculated from the fitting of the measured linear tails together with the blueshift (as compared to the single-particle dispersion shown by the light-blue dashed lines). Full temporal dynamics of the PL vs. $k_y$ is provided in the Supplementary Movie 1, while additional snapshots corresponding to different densities are shown in the SI.

Experimentally following this dynamics allows to observe in time the delayed formation of the dark condensate after the pulse arrival, accompanied by the theoretically predicted energy-flat parts of the excitation spectrum, its shifting down with time due to the decreasing blueshift and, finally, disappearance of the flat parts and recovering of the single-particle near-parabolic LLP dispersion as the system goes below threshold. Finally, Fig. 2f shows the energy-resolved PL image along the $k_x$–direction above threshold. While the BIC polariton condensate in the local minimum of the modified dispersion stays dark, the excitations on top of the condensate occupying the very narrow region around $k_x = 0$ are emitting light. Here, since under pulsed nonresonant excitation the blueshift is strongly time dependent, the resulting time-integrated measurement in Fig. 2f looks apparently broadened in energy (see the corresponding enlarged curves for different densities in the inset of Fig. 2c, getting smeared over time) which appears as a two-lobe image.

### Dispersion of collective excitations at different $k_x$ under continuous-wave excitation

In order to see a more pronounced linearisation of the dispersion tails, we turn to continuous-wave (c.w.) excitation that allows to reach a density state while avoiding the energy blur induced by the dynamical change of energy of the condensate under time-integrated measurements. For fixed excitation conditions, we compare the PL from energy dispersion versus $k_y$ resolved at different values of $k_x$ as shown in Fig. 3: one cut closer to $k_x = 0$ (a), another directly within the bright spot with maximal emission that is seen above threshold in Fig. 2f (b), and a final one when $k_x$ is further away from the center (c). As one can observe, while the PL from the non-condensate (out-of-the-BIC) lower polaritons in panels (a) and (c) is weak enough to allow registering the emission from the upper branch, in panel (b) the PL from the flat part of the dispersion is very bright, making the ULP invisible on the same (normalized) scale. Matching the observed PL with the theoretical expressions for the dispersion of excitations Eq. (3) and blueshift

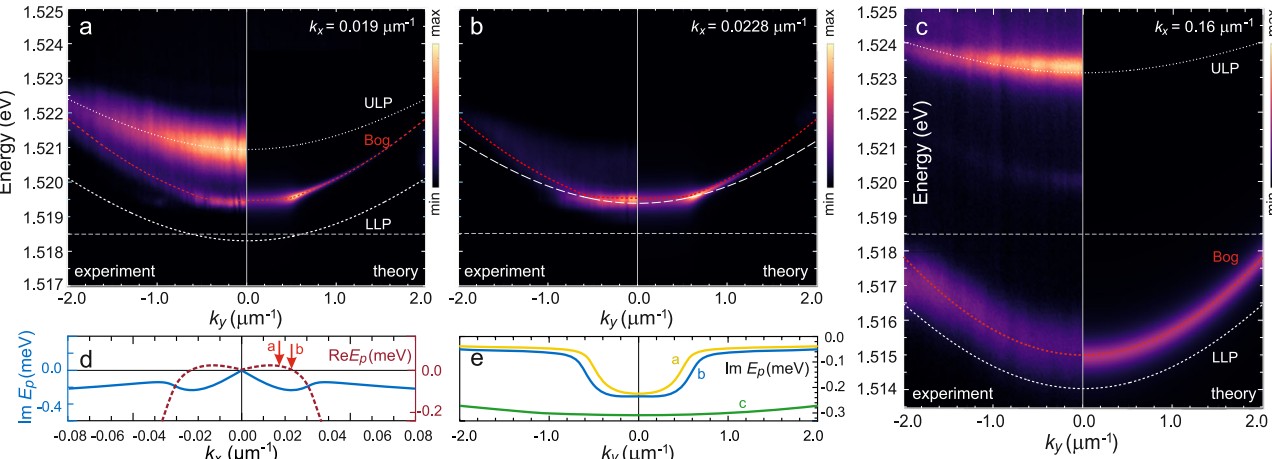

**Fig. 3 | Continuous-wave excitation study. a–c** Experimentally measured (left-hand sides) and theoretically calculated (right-hand sides) photoluminescence from the system vs. $k_y$ at three different values of $k_x$ above threshold: **a** $k_x = 0.019\ \mu m^{-1}$, **b** $k_x = 0.0228\ \mu m^{-1}$, **c** $k_x = 0.16\ \mu m^{-1}$. The overlaid red dotted lines represent the real part of the excitation spectrum $\mathrm{Re}E_\mathbf{p}(k_y)$ at corresponding values of $k_x$, white dotted lines in **a**, **c** show the ULP and LLP single-particle dispersions. In **b**, the blueshifted lower polariton dispersion is shown for comparison. Intensity colorscales are in arb. units and normalized for each panel separately. **d** Real part (dashed line), counted from the blueshifted saddle point of the dispersion, and imaginary part (solid line) of $E_\mathbf{p}$ vs. $k_x$ in the vicinity of $k_x$ - 0. The red arrows indicate the values of $k_x$ at which the panels **a** and **b** are measured. **e** Imaginary part of the spectrum vs. $k_y$ corresponding to panels **a** (yellow), **b** (blue), and **c** (green). For all panels, calculation parameters are the same as in Fig. 2 and densities estimated from fitting are $n_0 = 5 \times 10^{10}\ cm^{-2}, \bar{n} = 11 \times 10^{10}\ cm^{-2}$.

Eq. (2) for all three cases and using the condensate and reservoir densities as fitting parameters allows us to define exactly the values of $k_x$ at which those cuts were measured.

The first straightforward outcome of this study is the clearly increased slope of the linear part of the Bogoliubov dispersion for the states lying above the flat stripe, as shown in Fig. 3a, b. While the difference in energy at which the energy-flat part is seen in (a) and (b) lies within the linewidth (see the red arrows in Fig. 3d or the inset of Fig. 2c), the length of the stripe is changing with $k_x$ and is different for (a) and (b), in agreement with the theoretical Fig. 2b. The tails of the dispersion outside the flat part in both (a) and (b) are clearly more visible than in the case of pulsed excitation studied above, and are linearized in accordance with Eq. (3). The shifted LLP dispersion is plotted in panel (b) together with the Bogoliubov spectrum. When moving away from the condensate, the dispersion of excitations recovers the shape of the lower polariton branch, while still staying blueshifted, as shown in Fig. 3c. For comparison, in the right-hand sides of panels Fig. 3a–c we plot the theoretically calculated PL emission, using the excitation spectrum derived above (note that since we neglected the upper polaritons in the theory, we do not reproduce the ULP luminescence). In this calculation, we assume that at finite temperature the normal branch of the Bogoliubov spectrum is predominantly populated (due to thermal depletion of the condensate).

As a more subtle and more meaningful insight from fine resolving the PL in $k_x$ close to the BIC, we reveal that the maximal emission (panel b), i.e. the flat line in the Energy vs. $k_y$ dispersion, corresponds to the point in $k_x$ where the real part of the excitation spectrum $E$ (counted from the blueshifted saddle point) goes to zero. This analysis is presented in Fig. 3d, where we indicate by arrows the two values of $k_x$ at which panels (a) and (b) were measured ($k_x$ corresponding to panel (c) lies out of the range of values plotted in (d) and is not marked). As discussed above, the only values of $k_x$ where $\mathrm{Re}E_\mathbf{p} = 0$ correspond to the points in which the imaginary part of $E_\mathbf{p}^2$ changes sign, i.e. the points that lie at the same energy as the dark condensate in the BIC, thus making the scattering from the condensate towards these states energetically effortless. Looking at the imaginary part of the dispersion, plotted as well in Fig. 3d, allows us to conclude that since $\mathrm{Im}E_\mathbf{p}$ is negative for all $k_x > 0$, the condensate with $\mathrm{Im}E_\mathbf{p} = 0$ stays nevertheless

stable and there is no macroscopic gain in the states indicated by the arrow 'b'. However, since they correspond to the largest $|\mathrm{Im}E_\mathbf{p}|$, these states correspond to the maximum PL intensity observed in the experiments. We conclude that the particles residing in the dark condensate at $k_x = 0$ at any scattering event may jump to the states 'b' where they leak out of the system due to maximal loss. For completeness of the analysis, we plot $\mathrm{Im}E_\mathbf{p}$ vs. $k_y$ in Fig. 3e. One can see that the large imaginary part and hence short lifetimes are characteristic only for the states along the energy-flat part of the dispersion $E_\mathbf{p}(k_y)$, whereas the linear tails, corresponding to low-lying excitations above the saddle-point condensate, feature narrow linewidth and can be described adequately within the equilibrium theory.

## Anisotropy in the momentum space

To study the anisotropic dispersion of excitations discussed above, we measure the photoluminescence emission under pulsed excitation in the plane $(k_x, k_y)$ which is shown in Fig. 4 for different energies close to the BIC state. For a clearer visualization, we suggest to watch the Supplementary Movie 2 showing the experimental cross-cut of the PL emission in the far field at different energies. The exotic shape with two parallel stripes in the middle at a given energy can be directly related to the populated states of the excitation spectrum where $\mathrm{Re}E_\mathbf{p} = 0$ and the absolute value of the negative $\mathrm{Im}E_\mathbf{p}$ is large, as discussed above and confirmed by the calculated $\mathrm{Im}E_\mathbf{p}$ dependence on $k_x$ and $k_y$ in Fig. 3d, e. It is important to note that the distribution of the PL in two parallel stripes appears only above threshold: without the macroscopic population of the saddle-point, there are no states lying at the same energy with the condensate and hence no population of the energy-flat states along $k_y$ that exhibit strong emission. For comparison, we provide in the SI and the Supplementary Movie 3 the corresponding data taken below threshold, where the energy corresponding to the saddle point stays fully dark. In the above-threshold case of Fig. 4, due to time averaging in the pulsed excitation setting, the two-stripe pattern which lies at the energy of the condensate is visible in more than one panel, as the condensate shifts down over time (due to decreasing blueshift). We note that the observed peculiar far field PL distribution in this case is very different from the two-lobe pattern of grating-based lasers[8,9], where, despite the similar dark line along $k_x = 0$, the lasing happens exclusively at the photonic BIC

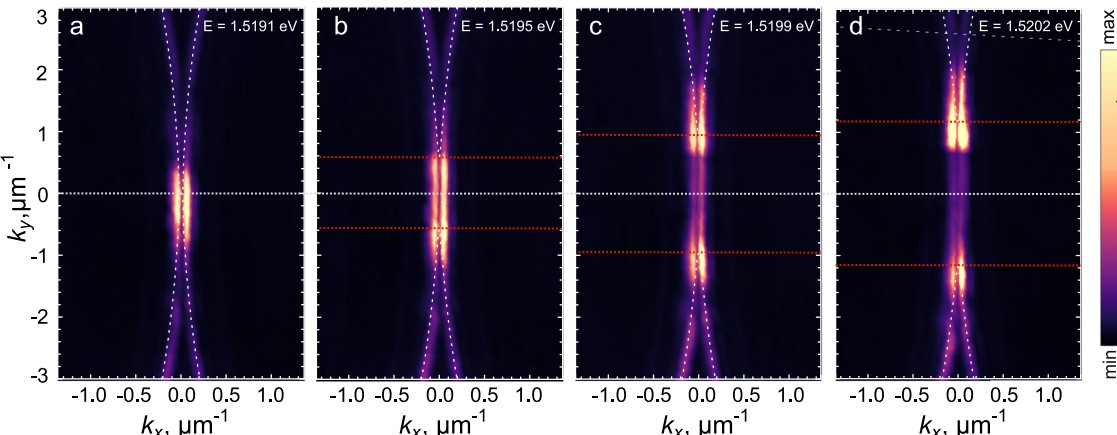

**Fig. 4 | Momentum-space scan. a–d** Experimental PL maps in the $(k_x, k_y)$ domain taken at different energies just above the blueshifted BIC polariton condensate (as denoted on the panels). The line $k_x = 0$ above threshold stays fully dark, making the PL profile in the vicinity of the condensate look like two parallel stripes corresponding to the flat parts of $\mathrm{Re}E_\mathbf{p}(k_y)$ appearing at small non-zero $k_x$. These maps are time-integrated in the pulsed excitation setting. Intensity colorscale is in arb. units and normalized the same way for all panels. Condensate and reservoir densities $n_0 = 5 \times 10^{10}$ cm$^{-2}$, $\bar{n} = 4 \times 10^{10}$ cm$^{-2}$, other parameters as in Fig. 2.

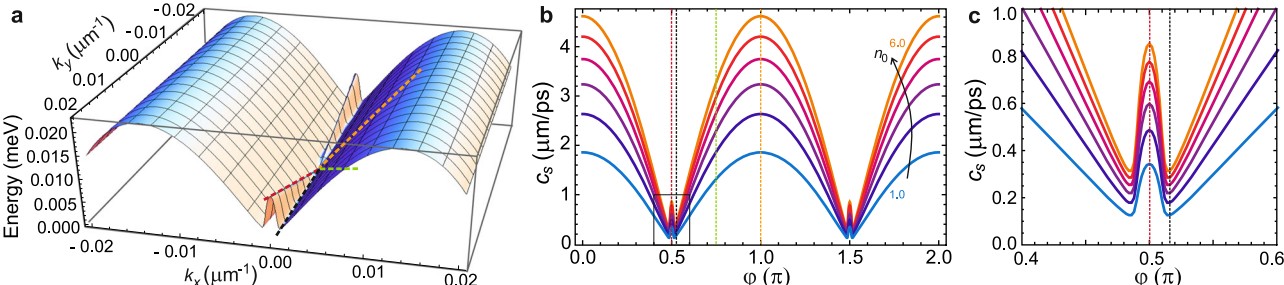

**Fig. 5 | Anisotropic Bogoliubov sound velocity. a** A close-up view of the local potential well created in the real part of the Bogoliubov spectrum of elementary excitations around the saddle point, with essentially different slopes $c_s^x \gg c_s^y$ (where $c_s^x = c_s(\varphi = \pi m), c_s^y = c_s(\varphi = \pi[m+1]/2), m \in \mathbb{Z}$). The colored dashed lines are guide to the eye showing different directions along the anisotropic dispersion slopes linearized in the vicinity of $k = 0$. **b** The Bogoliubov sound velocity profile according to Eq. (4). The colored dashed lines mark different angles corresponding to the slopes marked in **a**, see discussion in the text. **c** Close-up of the sound velocity profile at the vicinity of $\pi/2$ (corresponding to the black rectangle in **b**). The condensate density $n_0$ in **b, c** is varied from 1 to $6 \times 10^{10}$ cm$^{-2}$, $\bar{n} = 4 \times 10^{10}$ cm$^{-2}$, other parameters as in Fig. 2.

energy and the intensity distribution along $k_y$ is a Gaussian centered at $k_y = 0$. In our case, the PL emission along both directions and in energy follows the calculated dispersion Eq. (3), with the length of the parallel lines shorter when the condensate density is lower, and the curved tails at nonzero momenta appearing outside the two-stripe region (whose emission is weaker since $|\mathrm{Im}E_\mathbf{p}|$ is much smaller). In all panels of Fig. 4, the theoretically calculated cross-cuts of the real part of $E_\mathbf{p}$ at the corresponding energies above the BIC state are overlaid as dashed lines on top of the experimental images. The maximum of the experimental emission appears at those states where the real part of the dispersion $E_\mathbf{p}$ crosses the given energy (the brighter spots move away from the center in the panels (c) and (d) that correspond to higher energies).

From the point of view of the anisotropy of collective excitations which is illustrated by the above study of the spectrum, it is also interesting to address the limit of very small momenta. Experimentally, since the region of $k_x$ corresponding to the positive slope of $\mathrm{Re}E_\mathbf{p}$ is very small, the shift in energy which is acquired within this narrow $\Delta k_x$ lies within the linewidth. At the same time, it is not possible to follow the change in energy with $k_y$ along the positive slope of the saddle (from Fig. 2a one sees that at $k_x \to 0$ the energy-flat part along $k_y$ disappears and the dispersion is linear) due to the darkness of the states corresponding to $k_x = 0$ (at any $k_y$). Theoretically, however, taking the limit $p \to 0$ in Eq. (3) allows to obtain the sound velocity which appears to be strongly dependent on direction (on the angle $\varphi$ in the polar

coordinate system):

$$c_s(\varphi) = \mathrm{Re}\left\{ \sqrt{ \mu_{\mathrm{LP}}\left[ \cos^2\varphi\left(\frac{1}{m_{\mathrm{LP}}^x} + \frac{i}{s_x}\right) + \frac{\sin^2\varphi}{m_{\mathrm{LP}}^y}\right]\left[1 + \frac{2\mu}{\sqrt{(\hbar\Omega)^2 + \Delta^2}}\right] } \right\},$$

$$(4)$$

where $m_{\mathrm{LP}}^{x(y)}$ are the effective masses of LLP in the $x(y)$–direction, and $s_x$ is the mass-like parameter defined by the radiative losses rate $\gamma$ and the radiative coupling of the modes $U$ (for more details, see the SI). In a standard case of real positive masses in both directions, for experimentally-relevant parameters (resulting in $1/|m_{\mathrm{LP}}^x| \gg 1/m_{\mathrm{LP}}^y$) the velocity given by Eq. (4) would significantly increase when approaching the $x$–axis (the direction corresponding to $\varphi = \pi m, m \in \mathbb{Z}$). However, here the expression for the sound velocity is modified by the negative $m_{\mathrm{LP}}^x$ and by the imaginary contribution $s_x$, which makes the definition of the sound velocity more complicated. In Fig. 5b we plot the sound velocity dependence on the angle $\varphi$ revealing a very anisotropic profile. The increase of $c_s$ in the $x$–direction is clearly seen from the difference of slopes in the linearized parts of the Bogoliubov dispersion (see the zoom-in at the low-momenta region in Fig. 5a): while the slope along $k_y$ at $k_x = 0$ corresponds to very low $c_s^y$, the shallow parts of the spectrum at finite but small $k_x$ result in the sound velocity in the directions slightly deviating from the $y$–axis being very

close to zero (see Fig. 5c). Much steeper slopes along $k_x$ at $k_y = 0$ provide maxima of $c_s$ at $\varphi = \pi m$, as shown in Fig. 5b for various condensate densities. It is important to note that taking into account the non-radiative exciton losses would additionally affect the behavior of the excitation spectrum along $k_x$ in a very narrow vicinity of the saddle point. A more detailed discussion of this matter is provided in the SI.

## Discussion

In this work, we studied theoretically and experimentally the spectrum of elementary excitations of the BIC polariton condensate arising from the coupling of excitons to the photonic modes in a patterned semiconductor waveguide. The studied system is exotic and very different from the well-studied case of microcavity polaritons. We show that despite the saddle-like shape of the single-particle dispersion of lower polaritons, which exhibits a maximum in one of the directions in $k$−space, with the accumulation of particles in the saddle point, a local minimum is created with the positive slopes of the Bogoliubov dispersion, which may help in further accumulation of polaritons. At the same time, two parallel stripes in momentum space, which correspond to the states at the same energy as the condensate—but being bright—appear. It is, in fact, the interplay of the negative mass and the momentum-dependent losses that results in the existence of such unique, extended in momentum, states outside the condensate. This unusual anisotropic shape of the excitation spectrum yields in the $k_y$−direction to two energy-flat zones (near $k_x = 0$) separated by the dark line at $k_x = 0$. Thanks to the quasi-infinite lifetime of the BIC state, the condensate itself does not emit light hence allowing to neatly observe the populated states around and above the saddle point. With both the slopes of the linear parts of the dispersion that follow the energy-flat regions and the interaction-induced blueshift being directly experimentally accessible, our theory allows to precisely define the macroscopic density of the dark polariton condensate. Thus, while the observation of the excitation spectrum in such a system is striking on its own, it also allows to obtain the information about the saddle-point polariton condensate which is otherwise invisible.

Remarkably, despite the negative mass in the $x$−direction and the presence of losses, the real part of the spectrum is positive everywhere at $k \to 0$, allowing to extract the velocity of sound in every direction. The latter appears to be highly anisotropic, dropping to almost zero in the directions next to the $y$-axis, i.e. slightly misaligned with the grating principal axis, and growing in the direction of propagation along the waveguide. It needs to be noted that the anisotropic sound velocity was reported previously for atomic BECs with anisotropic dipolar interactions[31,32]. In case of dipolar BECs, however, transport measurements reveal the anisotropy of the critical Landau velocity rather than the velocity of sound[33], since the dispersion of excitations is affected by direction-dependent roton softening. The situation realized in our work can be better compared with anisotropic superfluidity reported for excitons[34] and cavity photon BEC[35] in periodically modulated planar structures. At the same time, we stress that here we deal with a more exotic case of anisotropy: even though there is a well-defined finite sound velocity, superfluidity in the sense of Landau criterion cannot be reached in all directions of propagation. Indeed, no matter how small the velocity would be below $c_s$ in the $x$−direction, due to the negative slopes of the dispersion, it could always elastically create an excitation out of the condensate. However if an obstacle is propagating strictly along the $y$−axis, the velocity $c_s^y$ would be a proper Landau sound velocity below which superfluidity can be observed. This illustrates the richness of saddle-point condensates and their anisotropic behaviors.

Along with the previously proposed topological-dispersion engineering[36], we believe that this work demonstrates the high interest in engineering the excitation spectrum of the condensate e.g. via implementing different grating symmetries—as a tool to impart new

properties to the condensate itself—and underlines once again the astonishing richness of polariton systems, as well as multiple possible avenues for further investigations.

## Methods

### Theoretical methods

Starting from the Hamiltonian (1) in the neglection of the ULP fields, one can separate the macroscopically occupied condensate state as

$$\hat{P}_-(\mathbf{r},t) = \sqrt{n_0} + \hat{P}'_-(\mathbf{r},t), \quad \hat{Q}(\mathbf{r},t) = X_0\sqrt{n_0} + \hat{Q}'(\mathbf{r},t), \tag{5}$$

where $n_0$ is the condensate density, $\hat{Q}'(\mathbf{r},t) = \int X(\mathbf{r}'-\mathbf{r})\hat{P}'_-(\mathbf{r}',t)d\mathbf{r}'$, and the average $\langle\hat{P}'_-(\mathbf{r},t)\rangle = \langle\hat{Q}'(\mathbf{r},t)\rangle = 0$. The substitution Eq. (5) allows to rewrite the triple products of the field operators in Eq. (1) in the Hartree-Fock mean-field approximation as

$$
\begin{aligned}
\hat{Q}^\dagger(\mathbf{r},t)\hat{Q}(\mathbf{r},t)\hat{Q}(\mathbf{r},t) = {} & X_0\sqrt{n_0}(|X_0|^2 n_0 + 2n'_Q) \\
& + 2\hat{Q}'(\mathbf{r},t)(|X_0|^2 n_0 + n'_Q) + \hat{Q}'^\dagger(\mathbf{r},t)X_0^2 n_0,
\end{aligned} \tag{6}
$$

$$\hat{\tilde{Q}}^\dagger(\mathbf{r},t)\hat{\tilde{Q}}(\mathbf{r},t)\hat{Q}(\mathbf{r},t) = \left[X_0\sqrt{n_0} + \hat{Q}'(\mathbf{r},t)\right]\tilde{n} \tag{7}$$

with notations $n'_Q = \langle\hat{Q}'^\dagger(\mathbf{r},t)\hat{Q}'(\mathbf{r},t)\rangle, \tilde{n} = \langle\hat{\tilde{Q}}^\dagger(\mathbf{r},t)\hat{\tilde{Q}}(\mathbf{r},t)\rangle$. Using the Hamiltonian Eq. (1) with the substitutions Eqs. (5) and (6), (7), the Heisenberg equation for the non-condensed part of the polariton field $\hat{P}'_-(\mathbf{r},t)$ can be written as

$$
\begin{aligned}
i\hbar\partial_t \hat{P}'_-(\mathbf{r},t) = {} & [\varepsilon_-(-i\hbar\nabla) - \mu_-]\hat{P}'_-(\mathbf{r},t) \\
& + g\int d\mathbf{r}' X^*(\mathbf{r}'-\mathbf{r})\left[(2n'_Q + 2n_0|X_0|^2 + \tilde{n})\hat{Q}'(\mathbf{r}',t)\right. \\
& + \left. X_0^2 n_0 \hat{Q}'^\dagger(\mathbf{r}',t)\right].
\end{aligned} \tag{8}
$$

In Fourier space, (8) takes the form $i\hbar\partial_t \hat{P}_\mathbf{p}(t) = [\varepsilon_-(\mathbf{p}) - \mu_- + g(2n_0|X_0|^2 + 2n'_Q + \tilde{n})]|X_\mathbf{p}|^2\hat{P}_\mathbf{p}(t) + gn_0 X_0^2 X_\mathbf{p}^{*2}\hat{P}_{-\mathbf{p}}^\dagger(t)$ that allows to perform the Bogoliubov transformation

$$\hat{P}_\mathbf{p}(t) = u_\mathbf{p}\hat{\alpha}_\mathbf{p} e^{-iE_\mathbf{p}t/\hbar} - v_{-\mathbf{p}}^*\hat{\alpha}_{-\mathbf{p}}^\dagger e^{iE_\mathbf{p}^*t/\hbar} \quad \text{with } |u_\mathbf{p}|^2 - |v_{-\mathbf{p}}|^2 = 1, \tag{9}$$

where $\hat{\alpha}_\mathbf{p}$ is the annihilation operator of the Bogoliubov excitation with momentum $\mathbf{p}$. Eq. (9) yields the excitation spectrum Eq. (3). We note that due to the underlying photon dispersion whose imaginary part is strongly momentum-dependent, the Hopfield coefficients $X_\mathbf{p}, C_\mathbf{p}$ as well as the Bogoliubov amplitudes $u_\mathbf{p}, v_{-\mathbf{p}}$ are complex, and careful treatment of complex conjugations throughout the theory is required. Moreover, choosing the Bogoliubov transformation in the specific shape given by Eq. (9) defines the so-called ghost branch (GB) of excitations as $E_\mathbf{p}^{GB} = -E_\mathbf{p}^*$ (this definition of the GB is consistent with that used in ref. 27).

Theoretical calculations of the intensity of photoluminescence from the excitations around the BIC state that are presented in Fig. 4 are performed using the standard expression[37,38]:

$$\text{PL}(k_x, k_y; \omega) \propto \left(1 - |X_\mathbf{p}|^2\right)\text{Re}\int_0^\infty dt\, e^{-it(\omega - i0^+)}\langle P_\mathbf{p}^\dagger(t)P_\mathbf{p}(0)\rangle. \tag{10}$$

### Experiments

All the PL data reported in this work has been acquired using a confocal setup. The objective back focal plane is imaged on the entrance slit of the spectrometer allowing to obtain the energy versus $k$ dispersion. Two different magnification of the objective focal plane are used in order to properly image the dispersions along $k_x$ and $k_y$ which have very different effective masses and hence very different extension in the $k$-space. The laser used in the experiments is a fs-pulsed laser

having a repetition rate of 80 MHz and a pulse duration of ~100 fs. Excitation wavelength is set at 780 nm. The time resolved measurements are obtained by scanning the far-field emission and acquiring for each $k_y$ an energy versus time temporal trace with a streak camera. The set of temporal traces is then used to reconstruct the dynamics in the Energy vs $k_y$ space. Supplementary Movie 1 shows an example of temporal dynamics reconstructed in this way. The emission in the $(k_x, k_y)$–plane at a given energy like the images shown in Fig. 4 are obtained by scanning the far-field emission and acquiring a set of Energy vs $k_x$ dispersions, each one corresponding to a different $k_y$. The data are then merged and cut at a given energy in the plane $k_x, k_y$. Supplementary Movies 2 and 3 show a sequence of such cuts for different energies above and below threshold, respectively.

## Data availability

Relevant datasets generated and/or analyzed during the current study are available in the Open Science Framework (OSF) repository under the link https://osf.io/8zu5f/?view_only=5d54939f8c584c598308244c 34c34348. Data and any other information are also available upon reasonable request.

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

## Acknowledgements

We are grateful to Paolo Cazzato for technical support. The authors acknowledge the financial support of the Russian Foundation for Basic Research (RFBR) Grant No. 20-52-7816 (joint with CNR) and the MEPhI Priority 2030 Program (A.G. and N.V.), the Foundation for the Advancement of Theoretical Physics and Mathematics "BASIS" (A.G.), the Italian Ministry of University (MIUR) for funding through the PRIN project "Interacting Photons in Polariton Circuits" – INPhoPOL (grant 2017P9FJBS), project "Hardware implementation of a polariton neural network for neuromorphic computing" – Joint Bilateral Agreement CNR-RFBR – Triennal Programm 2021-2023, and the PNRR MUR projects "Integrated Infrastructure Initiative in Photonic and Quantum Sciences" I-PHOQS (IR0000016) and "National Quantum Science and Technology Institute" NQSTI (PE0000023). We thank Scott Dhuey at the Molecular Foundry for assistance with the electron beam lithography. Work at the

Molecular Foundry was supported by the Office of Science, Office of Basic Energy Sciences, of the U.S. Department of Energy under Contract No. DE-AC02-05CH11231. This research is partly funded by the Gordon and Betty Moore Foundation's EPiQS Initiative, Grant GBMF9615 to L. N. Pfeiffer, and by the National Science Foundation MRSEC grant DMR 1420541.

## Author contributions

N.V., D.S. and V.A. initiated the research project; A.G. and N.V. performed the calculations and theoretical analysis; F.R. designed the grating structures, growth was performed by K.B. and L.P.; M.E.-T., V.A. and D.S. realized the experiments and analyzed the data together with N.V.; N.V., V.A. and D.S. drafted the manuscript, and all the authors, including M.D.G., D.T. and D.B., were involved in the discussion of results and the final manuscript editing.

## Competing interests

The authors declare no competing interests.
