## [Peer Review File · Nature Communications]

Collective excitations of a bound-in-the-continuum condensateREVIEWER COMMENTS

Reviewer #1 (Remarks to the Author):

This manuscript from A.M. Grudinina et al. reports on the observation of Bogoliubov excitations in a bound-in-the-continuum (BIC) polariton condensate, and the calculation of the speed of sound in this system. This work is based on the very good agreement between theoretical models and experimental data. The sample and the experimental data are of great quality.

Although the topic is interesting and could be suitable for publication in Nature Communications, I have the impression that this work is just an incremental follow up of the recent paper published in Nature (Ref.9 in the text) from the same groups. Bogoliubov excitations and the linearization of the dispersion is a known phenomenon that have been largely studied in polariton systems and condensates. In polariton systems, this effect was first observed in 2008 (Ref. 18 in the text). The authors should explain clearly what the BIC system adds to the physics that is already known. The authors state that the long-living nature of the BIC condensate allows them to observed collective excitations with more details. Clearly, this results in high quality spectra, but it is not clear what new physics about this phenomenon is unveiled. I think the authors should comment more on this aspect in order to publish this work in a journal with high impact factor.

The anisotropy of the sound velocity and the resulting asymmetric superfluid regime is discussed at the end of the paper. This is a new and interesting aspect that is peculiar to the dispersion of the BIC polariton condensate. However, this is only discussed theoretically and no real demonstration is provided. A proof of concept of this phenomenon and a detailed discussion of possible applications would make the manuscript much stronger.

In figure 3, the authors comment on the change of the PL distribution as a function of the energy. It is not straightforward to see the reason behind this observation. Maybe a more detailed discussion could be added to the text.

In conclusion, despite the research described in this manuscript is of good quality, I do not feel like recommending it for publication in Nature Communications. At the current state, the manuscript reads as incremental to the work recently published in Nature. However, I believe the authors could leverage more on the unique properties of BIC condensates to provide more insights on the physics of collective excitations and polariton condensates.

Reviewer #2 (Remarks to the Author):

The manuscript 'Collective excitations of a bound-in-the-continuum condensate' by Grudinina et al. is an experimental and theoretical work devoted to the bound-state-in-the-continuum polariton condensates. The authors study the grating microcavity system (a waveguide) under low-power excitation, and then, over-threshold excitation and check the properties of the spectrum in these regimes extracting the information regarding the bright and dark exciton densities and the sound Bogoliubov modes. The experiment is supported by the standard theory based on the Bogoliubov model and the Hopfield coefficients.

In the abstract the authors write, 'long-living condensates in thermal equilibrium'. What do the authors mean by thermal equilibrium? Is this statement accurate?

They write, 'their superconducting and collective properties...'. However, superconducting properties are also collective. This makes this statement inaccurate.

The theory allows to define the macroscopic density of the dark polariton condensate. What are the benefits of that finding?

Why is the information regarding the saddle-point polariton condensate important? And also, what is the importance of knowing the sound velocity of the Bogoliubov modes in different directions?

While the results of the authors look solid, it is difficult for me to expect a broad interest of this manuscript, thus, I believe the manuscript is more suitable for a different journal.

Reviewer #3 (Remarks to the Author):

Grudinina and coauthors present a study on the excitation spectrum of a polariton condensate in a periodically modulated waveguide geometry that gives rise to a bound state in the continuum (BIC). The BIC is topologically protected from radiative losses, but may couple to excitons, resulting in polaritons with nominally infinite lifetime, which may also form a condensed state.

In the present manuscript the authors utilize this lack of coupling to the outside world to investigate the excitation spectrum of the condensate. Typically, this is a difficult task as the excitation spectrum yields a weak signal forming on a strong background signal arising from the condensate. Here, the condensate is essentially non-radiative which renders it easy to investigate the excitation spectrum.

The manuscript can be divided into an experimental and a theoretical part. The experimental part is the direct observation of the excitation spectrum of a nontrivial condensed state, which is certainly a novel and interesting result. The authors have taken great care to ensure the validity of their experimental results. Supplementary video 1 is a testimony to this as the authors have taken great care to

demonstrate that they can remove any inhomogeneous broadening due to temporal averaging from their data. However, the experimental part also shows some shortcomings where the authors might want to explain their results a bit better. I noticed especially the following points:

- Some of the figures shown in the manuscript lack critical information. For example, almost all the experimental photoluminescence plots lack information about how the color scale translates into intensities. It is also not clear whether the plots use a linear or logarithmic scale. The authors should clarify this.

- In figure 3 and supplementary videos 2 and 3 the authors show fully momentum-resolved cuts through the dispersion at individual energies close to the blueshifted BIC state. Unfortunately, these plots confuse me heavily. Judging from figures 1,2 and S3, the blueshifted BIC condensate forms at an energy of about 1.519 eV. The caption of figure 3 says that the calculation parameters are identical to figure 1. The lowest energy shown in figure 3 is 1.5322 eV, which is more than 13 meV above the nominal BIC state. Judging from figure S1, this spectral region should correspond roughly to the gap between the upper lower polariton branch and the upper upper polariton branch. Can the authors elaborate?

- The credibility of the results depends heavily on whether the authors can convincingly demonstrate that the dispersion they measure actually really follows the theoretical one, both in terms of being flat at very small momenta and becoming linear for slightly larger momenta. I tend to believe the authors that they actually achieved this, but it is not actually trivial to see this from their data. The spectra shown in figure 2 seem to be flat, but it might be beneficial for the reader if the blueshifted standard parabolic dispersion was also shown for comparison, so that it becomes easier for the reader to spot the differences. The linearization of the dispersion is even harder to spot. There are some hints of linearization in the supplementary videos, but it is not trivial to see this due to the limited range of momenta, from which PL is actually observed. Is the PL intensity of the PL spectra shown on a linear or log scale? Potentially switching to a log scale might make it easier for the reader to see more of the dispersion.

The main theoretical contribution is the development of an analytical approximation to the shape of the Bogoliubov spectrum, its PL intensity and an angle-dependent expression for the sound velocity. In the theory section, the authors also emphasize some finer details of the excitation spectrum appearing at very small momenta and point out some things which render the anisotropic sound velocity interesting, e.g., directional superfluidity. I am not fully convinced by these discussions and the motivations presented by the authors. Some of my considerations are listed below:

- The authors place some emphasis on the discussion of sound velocities. Indeed these are critical for the question of superfluidity. However - as the authors correctly state - the negative dispersion along the x-direction will effectively prevent the system from showing superfluid properties. Along the same lines, the authors emphasize that this asymmetry may also be a feature rather than a bug: for propagation strictly along the y-axis, superfluid properties may arise. I am not entirely convinced by the claim and the authors might want to back it up. In the SOM, the authors discuss that the effects of non-radiative losses become apparent only in a narrow region in k which would require spot sizes of several hundreds of micrometers to create detectable effects. To me it seems like realizing flow pointing strictly along the y-direction will require similar spot sizes in order to ensure that only a narrow range in k -space

is populated. How does this compare to the spatial dimensions of their sample? The authors do not give the exact dimensions, but in the 2022 Nature paper on polariton BEC from a BIC written by a part of the current authors, the grating dimensions were 300 micrometers x 50 micrometers, where the smaller direction corresponds to the y-axis. Requiring a spot size larger than the actual dimensions of the sample does not seem like a realistic proposal to me. Realizing some kind of flow along this direction also seems quite non-trivial to me. For the same reasons, it is not clear to me whether the fine structures at low momentum which the authors focus on are really relevant or require unrealistically large system sizes to become observable.

- Along similar lines, asymmetric sound velocities and superfluid properties certainly are very interesting. However, asymmetric superfluid properties have of course been found in other systems as well and it would certainly be interesting to compare the authors' results to these other systems. Perhaps one of the most prominent examples of anisotropic superfluids is given by dipolar quantum gases, e.g., Physical Review Letters 121, 030401 (2018).

-The authors' statement that "we believe that this study could open a new way of controlling the condensate properties by engineering its excitation spectrum" is a bit vague. Do they have a concrete plan on how to engineer the excitation spectrum and how to utilize this to control the condensate? Which properties do they intend to control and why?

In summary, polariton BECs in periodically modulated waveguide geometries featuring BICs are still a relatively young topic and therefore a study of their excitation spectra is certainly relevant to the field - even although the results more or less clearly show that superfluidity will not be an issue here. Still, the saddle point condensation geometry will likely lead to interesting physics in the future. However, as pointed out above, partially the analysis seems a bit shaky and might benefit from a more thorough discuss and a clearer presentation of the data at hand. Also, some of the outlooks used for motivation might give the impression of being slightly overselling the results - also as outlined above. It is my impression that a revised manuscript that improves the explanation and presentation of the results and thereby makes it clearer that the evidence for the postulated claims is sound and also chooses a more moderate and realistic tone in terms of the outlook and the potential further research directions arising from the present work may be a reasonable addition to the literature and also a manuscript that reaches the standards of Nature Communications. However, I have doubts that the current revision of the manuscript meets these standards. Accordingly, my suggestion would be to reconsider a revised manuscript.

Some minor points:

- On page 3, discuss Fig. 1c and d where the PL emission below threshold is shown and state that the upper of the two polariton levels shown is more populated than the lower one. Are the authors sure about that? I would expect the upper levels to be extremely leaky, while the lower levels should be rather long-lived, so I am not sure that the magnitude of the PL directly reflects the magnitude of the population. Or does the lifetime already vary drastically along the lower branch?

- On page 3 of the supplementary materials some parts seem to be missing. In the right column, the sentence starting directly beyond the only equation in the column starts with "equations one sees[...]" which seems to indicate that the sentence is incomplete.

Authors' response to all comments of the Reviewers

GENERAL CONSIDERATION

We would like to thank all the Reviewers for carefully considering our work and finding ‘*both the experiment and theory of a great quality*’ and ‘*with very good agreement*’, ‘*the results solid*’ and ‘*the topic itself novel and interesting*’.

Before providing the point-by-point answer to all the Reviewers' questions and concerns (see below), we would like to generally comment on the statement appearing in some of the reports that this work may be “just a follow-up” from the previous paper published in Nature [current Ref. 12 in the revised manuscript bibliography].

After Bose condensation in a system of microcavity polaritons was demonstrated in 2006 [Nature **443**, 409 (2006)], the first study of its collective excitation spectrum was published in Nature Physics [S. Utsunomiya et al., Nat. Phys. **4**, 700 (2008)], followed by several Nature Communications publications [P. Stepanov et al., Nat. Commun. **10**, 3869 (2019); D. Ballarini et al., Nat. Commun. **11**, 217 (2020); M. Pieczarka et al., Nat. Commun. **11**, 429 (2020)], as well as many other journal papers studying various details of the polariton Bogoliubov spectrum of the same system. This is to no surprise, as the dispersion of excitations provides essential knowledge about the quantum and thermal condensate depletion, occupation of low-lying collective states, superfluidity etc.

Here we address polaritons in a system which, quoting one of the reviews, is a very young topic, with their condensation happening in a nontrivial state which is optically dark, has a negative effective mass and very peculiar properties. The current work is a first study of the collective excitations of such a condensate, and it addresses the excitation spectrum both experimentally and theoretically in great detail. Therefore, we definitely do not agree with the comment that, because the work follows the evidence of Bose condensation in a BIC state, it may be considered as “incremental”.

Nevertheless, as suggested by the Reviewers, in the revised version of the paper we have added more analysis, both experimental and theoretical, as well as requested discussion, and we believe that it answers all the raised concerns and renders the paper suitable for publication in Nature Communications.

Authors' response to remarks of Reviewer #1

This manuscript from A.M. Grudinina et al. reports on the observation of Bogoliubov excitations in a bound-in-the-continuum (BIC) polariton condensate, and the calculation of the speed of sound in this system. This work is based on the very good agreement between theoretical models and experimental data. The sample and the experimental data are of great quality. Although the topic is interesting and could be suitable for publication in Nature Communications, I have the impression that this work is just an incremental follow up of the recent paper published in Nature (Ref.9 in the text) from the same groups. Bogoliubov excitations and the linearization of the dispersion is a known phenomenon that have been largely studied in polariton systems and condensates. In polariton systems, this effect was first observed in 2008 (Ref. 18 in the text). The authors should explain clearly what the BIC system adds to the physics that is already known.

We thank the Reviewer for assessing our experimental data to be of a great quality and the agreement with the theory being very good, as well as the topic interesting and suitable for publication in Nature Communications.

Indeed, as the Reviewers rightly notices, the Bogoliubov spectrum in microcavity polariton systems and its linearization were studied in many details and these studies were published first in Nature Physics (current Ref. 21), then several Nature Communications (current Refs. 24, 25, 28), along with many other papers. None of these papers studying the excitation spectrum of microcavity polaritons were found to be “incremental follow-ups” of the observation of polariton Bose condensation in microcavities (current Ref. 14).

Here, we study polaritons in a waveguide geometry with grating featuring a BIC state. The polariton Bose condensate that was observed in Ref. 12 (Ref. 9 in the previous version) has therefore a negative effective mass in

one of the two spatial directions, is very anisotropic and optically dark. The dispersion of lower polaritons in this system is saddle-shaped. This system is completely different from most polariton systems (in which polaritons have a near-paraboloid dispersion in the region of small k). We would like to stress that polaritons showing such a saddle-like dispersion are, until now, basically unexplored.

The present work offers the detailed study of the spectrum of elementary excitations of the nontrivial dark BIC condensate in such a saddle point. We show that the Bogoliubov spectrum consists of two elongated parallel energy-flat stripes around the condensate, while its linearization for the k_y -direction starts at some non-zero value of momentum. Due to these reasons we have to disagree with the Reviewer that we report on a “well-known phenomenon that has been largely studied”. The system that we consider here is very new and not studied at all compared to “regular” microcavity polaritons.

The authors state that the long-living nature of the BIC condensate allows them to observe collective excitations with more details. Clearly, this results in high quality spectra, but it is not clear what new physics about this phenomenon is unveiled. I think the authors should comment more on this aspect in order to publish this work in a journal with high impact factor.

We thank the Reviewer for praising the high quality of the observed spectra.

In fact, the observed emission of the BIC condensate in k -space has a peculiar shape of two parallel stripes continuing into curved tails (see Fig. 2f, and corresponding the new Fig. 4, previously Fig. 3). Interestingly, the two-lobe shape of far-field emission has been previously observed in lasing systems, for example perovskite-based surface-emitting lasers [see Adv. Mater. Technol., 1700253 (2018); ACS Nano **15**, 7286 (2021), the new Refs. 8, 9] where it arises as a Fourier transform of the near-field characteristics of the asymmetric lasing mode. Here, the “two-lobe” pattern appearing above the condensation threshold (due to nonlinearity of the system, see for example supplementary videos) is much more unique. Whereas in the case of lasers the emission happens exclusively at the energy of photonic BIC and does not feature any details along k_y , in the case studied here the PL from these two stripes is defined by the occupation of the states along the collective excitations dispersion (i.e. can be resolved in energy) and has a “fine structure”. With the new analysis that we added to the revised manuscript (new version of Fig. 4 and corresponding theoretical discussion in page 5), one sees that, first, the length of the parallel stripes along k_y is directly dependent on the condensate density, and second, that the intensity of photoluminescence along the two stripes is changing when resolved in energy. (Please see below a more detailed answer to the question about this Figure itself).

Figure 4 [revised version of previous Fig. 3]

The almost flat (in energy) part of the excitation spectrum obtained in this work does not follow from losses but from the symmetry features. Furthermore, we have performed new measurements with c.w. excitation changing the observation from snapshots to steady state. In the completely new Results subsection in page 4, in the added theoretical discussion we clarify that the losses actually stabilize the excitations resulting in the positive real part of the spectrum at small k_x . In the new Figure 3, we show that the energy-flat (versus k_y) parts of the spectrum correspond to the states which lie at the same energy as the condensate (see panel **d** of the figure below, red arrow tagged ‘b’), and indeed these are the states with the highest imaginary part and strongest observed PL. At the same time, the linearization of the spectrum is strongly dependent on direction in k -space. As we stress in the discussion added to the revised version of the manuscript, this shape directly follows the theoretically calculated shape of the Bogoliubov dispersion, with the almost-zero slope along the two stripes parallel to k_y and a more pronounced curvature outside the two-lobe region (see panels **a**, **b** in the Figure below).

Figure 3 [NEW]

Clearly, this physics is new and have not been previously reported neither for lasers, nor for polariton condensates. We agree with the Reviewer that the original version of the manuscript lacked a clear discussion of this novelty, which we amended in the revised version, providing both new measurements and a more extended theoretical investigation.

The anisotropy of the sound velocity and the resulting asymmetric superfluid regime is discussed at the end of the paper. This is a new and interesting aspect that is peculiar to the dispersion of the BIC polariton condensate. However, this is only discussed theoretically and no real demonstration is provided. A proof of concept of this phenomenon and a detailed discussion of possible applications would make the manuscript much stronger.

We are glad to know that the Reviewer finds the discussion of sound velocity and the resulting anisotropic superfluidity in the context of the investigated collective excitation spectrum interesting and peculiar. The proof of concept of varying sound velocity with changing the direction is in fact the excitation spectrum itself, which is reported both theoretically and experimentally, also with the increased resolution in momentum along k_x in the newly added experimental results.

To test anisotropic superfluidity in this case (BIC only formed at the Γ -point and therefore with zero group velocity) would require moving a defect against the resting BIC condensate, which is experimentally unfeasible. Alternatively, one could try to create the BIC state off the Γ -point of the folded bands, which would lead to a parametric rather than topologically protected BIC and requires a different kind of sample with different grating properties. These experiments are clearly beyond the scope of the current work. In the new version of the manuscript

we have revised the discussion of sound velocity so as to clearly state the observational limitations corresponding to the current experiment.

In figure 3, the authors comment on the change of the PL distribution as a function of the energy. It is not straightforward to see the reason behind this observation. Maybe a more detailed discussion could be added to the text.

We thank the Reviewer for this remark. In the revised version of the manuscript, we added the new, more detailed analysis, and updated prev. Fig. 3 (now Fig. 4) with the new version shown above. It displays, as before, the cuts of the k -space emission at different energies starting from the blueshifted condensate, but here overlaid with the contour cuts of the theoretical dispersion of excitations (which is shown in Fig. 2a). One sees that with the increase of energy the maximum of PL emission shifts along the parallel stripes from the center ($k_y = 0$) towards values $k_y > 0$. Since these images are time-integrated, one may see the two-stripe pattern appearing in the central region of all the energy cuts, however the PL intensity distribution in these images indicates the excitation states along the Bogoliubov dispersion which are occupied during the time of observation. The dashed lines are guide to the eye that explains the change of the PL. It is clearly seen here that the observed length of the flat part of the spectrum (the length of the two parallel stripes), as well as the curvature of ‘tails’ in the energy cuts very well coincides with the predicted spectrum. The maximum PL always lies in the region where the majority of allowed states on the dispersion at different k_x are very close to each other (see Fig. 2b). The corresponding discussion has been added to the text in page 5.

In conclusion, despite the research described in this manuscript is of good quality, I do not feel like recommending it for publication in Nature Communications. At the current state, the manuscript reads as incremental to the work recently published in Nature. However, I believe the authors could leverage more on the unique properties of BIC condensates to provide more insights on the physics of collective excitations and polariton condensates.

We once again are happy to read that the Reviewer finds our research of good quality and hope that we were able to underline in our response above the uniqueness of the observed excitation spectrum. We believe that the fact that the Bose condensation in this new, peculiar system of polaritons in the BIC has been reported in a previous work, does not make the study of collective excitations of such a dark condensate incremental at all. On the contrary, study of the excitation spectrum unveils new essential information about this rather unexplored system which, as we underline once again, is far from resembling microcavity polaritons. We think that we were able to strengthen the manuscript in its majorly revised version to clarify the concerns raised by the Reviewer, and that they find the paper now suitable for publication in Nature Communications.

Authors’ response to remarks of Reviewer #2

The manuscript ‘Collective excitations of a bound-in-the-continuum condensate’ by Grudinina et al. is an experimental and theoretical work devoted to the bound-state-in-the-continuum polariton condensates. The authors study the grated microcavity system (a waveguide) under low-power excitation, and then, over-threshold excitation and check the properties of the spectrum in these regimes extracting the information regarding the bright and dark exciton densities and the sound Bogoliubov modes. The experiment is supported by the standard theory based on the Bogoliubov model and the Hopfield coefficients.

First of all, we would like to thank the Reviewer for their interesting comments (that we answer below) that led to a better presentation of our results, improved theory discussion and new experimental data which, we believe, all helped to reach a much stronger version of the manuscript.

Here, we underline that our system is actually very different from the more common microcavity in which mode confinement and finesse are usually uneasy to optimize together. The electromagnetic modes that are strongly coupled

to the excitonic transition are confined by total internal reflection inside a waveguide, with minimal volume without the need for thick dielectric mirrors or other reflecting (often lossy) elements.

In particular, polaritons in a waveguide, thanks to clever engineering of the grating parameters can still be observed within the light cone yet with minimal losses. This is a great advantage compared to a standard microcavity structure. Moreover, the dispersion of lower polaritons in this system is saddle-shaped, which, again, is very different from microcavities (where polaritons have a near-paraboloid dispersion in the region of small k).

We would like to stress that the present system is completely new and until now basically unexplored in the context of polariton condensates. The present work therefore offers the first detailed study of the spectrum of elementary excitations of the nontrivial dark BIC condensate in such a non-parabolic and anisotropic dispersion.

Our second comment about the Reviewer’s assessment is that the theory used in the paper to derive the excitation spectrum is not at all a standard textbook Bogoliubov model. Its difference lies not only in weighting the interaction with the Hopfield coefficients, as the Reviewer suggests, but the theory contains also the consideration of finite temperature and the background reservoir density of dark excitons (both absent in the “standard” Bogoliubov approach even in its modification for polaritons [see e.g. Phys. Rev. Lett. **99**, 140402 (2007), Phys. Rev. B **85**, 075130 (2012), Phys. Rev. A **89**, 033626 (2014)]). As the Reviewer notes, diagonalization of the Hamiltonian averaged in the Hartree-Fock-Popov approximation allows us to distinguish the independent contributions to the excitation spectrum of the condensate itself, the thermal bright reservoir, and the dark exciton reservoir. We show that tracking these contributions separately allows to fit independently the condensate blueshift and the slopes of the linear parts of the dispersion in different directions, thus allowing to extract the condensate density and the reservoir density separately. We understand that these characteristics were not stressed enough in the previous version of the manuscript and, as the Reviewer pointed out, could have been more clearly stated. In order to better clarify the novelty of the theory, we comment on these differences to the standard cases in the revised manuscript on page 3.

Furthermore, in the revised version of the paper we have added the analysis of the theoretically-obtained excitation spectrum by addressing its imaginary part and discussing the role of the losses which here play the role of stabilizing the excitations which otherwise would not exist in the system. In particular, without the photon loss that provides the radiative (k -dependent) coupling of the counter-propagating modes and results in creation of the BIC itself, the seemingly-Hermitian Hamiltonian of the system would result, due to the negative mass in one of the directions, in a purely imaginary dispersion of excitations. Here, however, due to the presence of photon losses and the interplay of the negative mass with the imaginary part of the Hamiltonian, the resulting excitation spectrum has a non-zero and positive real part in the region of near-zero k_x (see new Figure 3).

Panels (d) and (e) from **Figure 3** showing the real and imaginary parts of E_p vs. k_x and k_y [NEW]

We added to the revised manuscript an extended discussion of both the real and imaginary parts of the dispersion, and their correspondence to the observed photoluminescence (see page 3 and the end of page 5).

In the abstract the authors write, 'long-living condensates in thermal equilibrium'. What do the authors mean by thermal equilibrium? Is this statement accurate?

The lifetime of polaritons is defined mostly by the photon lifetime since it is usually orders of magnitude shorter than the non-radiative lifetime of excitons. Approaching the BIC state, the lifetime of photons on the lowest branch of the dispersion goes to infinity. Thus the polariton lifetime in this state is quasi-infinite, limited only by the exciton non-radiative decay. In this regard the particles accumulated around the saddle point of the polariton dispersion are very much long-lived and have enough time to thermalize. This allows us to consider our system in thermal

equilibrium. At the same time, however, the losses play a crucial role in the formation of the observed elementary excitations dispersion (even while the condensate at $k = 0$ remains radiatively lossless). To avoid confusion, we removed the statement about thermal equilibrium from the revised Abstract.

They write, 'their superconducting and collective properties...'. However, superconducting properties are also collective. This makes this statement inaccurate.

We thank the Reviewer for pointing out this inaccuracy. We have amended it in the revised manuscript.

The theory allows to define the macroscopic density of the dark polariton condensate. What are the benefits of that finding?

Our findings allow to define the density of the state which is dark. There are no photons emitted, and hence there is no other way to assess the density of a condensate formed in the BIC state. Therefore obtaining this information from fitting the dispersion of excitations from such a condensate is essential. Moreover, even for the case of standard microcavity condensates the issue of extracting their density is quite well known since the usual technique is to work it out from the blueshift, which, however, is not always only dependent on the polariton density, but more often caused by the presence of dark and bright excitons in the reservoir.

Why is the information regarding the saddle-point polariton condensate important? And also, what is the importance of knowing the sound velocity of the Bogolibov modes in different directions?

As described previously, the condensate formed in the saddle-point of the polariton dispersion, in the BIC state, has a negative effective mass in one of the directions only, it is very anisotropic and optically dark. Despite the saddle point being a maximum rather than a minimum of the dispersion in one of the directions, the condensate formed there remains macroscopically occupied and exhibits very peculiar photoluminescence patterns (see e.g. Fig. 4). Thus investigating the spectrum of the system is needed even to explain the mere fact of stability of this saddle-condensate, as well as to provide information about occupations of the non-condensate states that explains the PL distribution. Low-energy collective excitations that we study here theoretically and experimentally provide direct information on the Bogoliubov sound in a quantum fluid, which is in turn critical for the thermodynamic and superfluid properties of this system. Propagation of sound is particularly interesting, since it is governed by Berezinskii-Kosterlitz-Thouless rather than Bose-Einstein condensation physics. Furthermore, collective excitations can reveal quantum corrections to classical symmetries, and quantum phase transitions. Given that the BIC polaritons and their Bose condensates are much different from those in microcavities, one can surely be interested in studying a system which is so novel and unexplored. Our work is the first study of elementary excitations from the bound-in-the-continuum condensate, thus paving the way towards future studies of all aforementioned physics in anisotropic systems with a negative mass.

While the results of the authors look solid, it is difficult for me to expect a broad interest of this manuscript, thus, I believe the manuscript is more suitable for a different journal.

We appreciate that the Reviewer finds our results solid and hope than now we were also able to convince them in the interest of this manuscript. In fact, we report the first observation of excitations from the intrinsically dark, asymmetric BIC condensate, with the PL showing two parallel stripes unveiling the energy-flat dispersion along the k_y -direction, that was never before observed or explained for a quantum fluid of interacting particles. We believe that the new analysis and more clear discussions added to the revised version of the manuscript, as well as the answers to all the Reviewer's questions, will make the Reviewer change their mind about the broad interest of the study and its suitability to be published in Nature Communications.

Authors' response to remarks of Reviewer #3

Grudinina and coauthors present a study on the excitation spectrum of a polariton condensate in a periodically modulated waveguide geometry that gives rise to a bound state in the continuum (BIC). The BIC is topologically protected from radiative losses, but may couple to excitons, resulting in polaritons with nominally infinite lifetime, which may also form a condensed state. In the present manuscript the authors utilize this lack of coupling to the outside world to investigate the excitation spectrum of the condensate. Typically, this is a difficult task as the excitation spectrum yields a weak signal forming on a strong background signal arising from the condensate. Here, the condensate is essentially non-radiative which renders it easy to investigate the excitation spectrum.

The manuscript can be divided into an experimental and a theoretical part. The experimental part is the direct observation of the excitation spectrum of a nontrivial condensed state, which is certainly a novel and interesting result. The authors have taken great care to ensure the validity of their experimental results. Supplementary video 1 is a testimony to this as the authors have taken great care to demonstrate that they can remove any inhomogeneous broadening due to temporal averaging from their data.

We thank the Reviewer for underlining the novelty and interest of our results, as well as praising the great care that was taken to finely resolve the experimental data to compare it to the theory.

However, the experimental part also shows some shortcomings where the authors might want to explain their results a bit better. I noticed especially the following points:

- Some of the figures shown in the manuscript lack critical information. For example, almost all the experimental photoluminescence plots lack information about how the color scale translates into intensities. It is also not clear whether the plots use a linear or logarithmic scale. The authors should clarify this.

All the photoluminescence plots in the current version of the manuscript are in linear scale. We agree with the Referee on the lack of information about the intensities in the previous version of the paper. To answer this point, we have added colorbars to each plot.

We also stress that we do not use the intensities to extract any quantitative information. For this reason the colorbars are in 'Arbitrary Units' or A.U., and are normalized to their maximum for a given panel or set of panels (as indicated now clearly in the revised versions of the Figure captions).

- In figure 3 and supplementary videos 2 and 3 the authors show fully momentum-resolved cuts through the dispersion at individual energies close to the blueshifted BIC state. Unfortunately, these plots confuse me heavily. Judging from figures 1,2 and S3, the blueshifted BIC condensate forms at an energy of about 1.519 eV. The caption of figure 3 says that the calculation parameters are identical to figure 1. The lowest energy shown in figure 3 is 1.5322 eV, which is more than 13 meV above the nominal BIC state. Judging from figure S1, this spectral region should correspond roughly to the gap between the upper lower polariton branch and the upper upper polariton branch. Can the authors elaborate?

We thank the Referee for spotting this mistake in the labels of the Supplementary Videos. We corrected the mistake and now the energy scales are consistent between all the Figures and Videos.

- The credibility of the results depends heavily on whether the authors can convincingly demonstrate that the dispersion they measure actually really follows the theoretical one, both in terms of being flat at very small momenta and becoming linear for slightly larger momenta. I tend to believe the authors that they actually achieved this, but it is not actually trivial to see this from their data. The spectra shown in figure 2 seem to be flat, but it might

be beneficial for the reader if the blueshifted standard parabolic dispersion was also shown for comparison, so that it becomes easier for the reader to spot the differences. The linearization of the dispersion is even harder to spot. There are some hints of linearization in the supplementary videos, but it is not trivial to see this due to the limited range of momenta, from which PL is actually observed. Is the PL intensity of the PL spectra shown on a linear or log scale? Potentially switching to a log scale might make it easier for the reader to see more of the dispersion.

We thank the Reviewer for this remark which prompted us to improve our work. In the revised version of the manuscript, we have added a new Figure 3 with new measurements which were made using a continuous wave pumping scheme to work in the steady state (compared to the time-resolved measurements which allowed us to track the dynamics and observe the excitation spectrum changing over time). In the new panels, we fine resolve the PL at different k_x close to zero and further away:

Panels (a,b) and (d,e) of **Figure 3** [NEW]

In panel (a), the cut is made at k_x lying still outside the bright stripe and closer to the dark region of the BIC (hence the PL intensity is weak and the upper branch is well visible), while panel (b) shows the cut at k_x directly along the bright stripe. The corresponding cut points are indicated in panel (d) by red arrows on the theoretical dispersion. In these new measurements, the flat part is wider in k_y and the linearization is much more pronounced due to higher densities of the condensate. In (b), we plot together with the Bogoliubov spectrum (red dotted line) the shifted single-particle lower polariton dispersion (white dashed line) so one can clearly see the difference of the slopes of the linear tails compared to the parabola. In fact, it is the slope of the tails combined with the length of the flat part that allows us to prove that what we evidence is the predicted Bogoliubov spectrum. We hope this to be more clear than the snapshot taken at different times during the time resolved experiments.

Switching to log scale, unfortunately, did not improve the visibility of the features, as the intensities along all the dispersions are relatively homogeneous (we remind the Reviewer that the condensate stays dark, while the observed PL is coming from the non-condensate states, this is why the upper branch, much less populated than the condensate, is often visible). Due to this reason switching to logarithmic scales does not improve the contrast. In the revised version of the paper, however, we did improve visibility of the data by switching to a slightly different color scheme. The new colorbars (given in a.u.) now appear on each image.

The main theoretical contribution is the development of an analytical approximation to the shape of the Bogoliubov spectrum, its PL intensity and an angle-dependent expression for the sound velocity. In the theory section, the authors also emphasize some finer details of the excitation spectrum appearing at very small momenta and point out some things which render the anisotropic sound velocity interesting, e.g., directional superfluidity. I am not fully convinced by these discussions and the motivations presented by the authors.

Along with the new measurements, in the revised version of the manuscript we provide as well the extended theoretical discussion of the behavior of the real and imaginary parts of the calculated Bogoliubov dispersion (shown in panels **d** and **e** of the new Fig. 3) which was not present in the initial version. This additional analysis allows us, in fact, to address in a much better way the states lying at the same level and above the saddle-point polariton condensate, as well as to explain the peculiar patterns observed in the PL emission. In the revised paper, therefore, the emphasis is very much shifted from the sound velocity discussion to the unique features of the excitations spectrum itself.

Some of my considerations are listed below:

- The authors place some emphasis on the discussion of sound velocities. Indeed these are critical for the question of superfluidity. However - as the authors correctly state - the negative dispersion along the x-direction will effectively prevent the system from showing superfluid properties. Along the same lines, the authors emphasize that this asymmetry may also be a feature rather than a bug: for propagation strictly along the y-axis, superfluid properties may arise. I am not entirely convinced by the claim and the authors might want to back it up. In the SOM, the authors discuss that the effects of non-radiative losses become apparent only in a narrow region in k which would require spot sizes of several hundreds of micrometers to create detectable effects. To me it seems like realizing flow pointing strictly along the y-direction will require similar spot sizes in order to ensure that only a narrow range in k -space is populated. How does this compare to the spatial dimensions of their sample? The authors do not give the exact dimensions, but in the 2022 Nature paper on polariton BEC from a BIC written by a part of the current authors, the grating dimensions were 300 micrometers x 50 micrometers, where the smaller direction corresponds to the y-axis. Requiring a spot size larger than the actual dimensions of the sample does not seem like a realistic proposal to me. Realizing some kind of flow along this direction also seems quite non-trivial to me. For the same reasons, it is not clear to me whether the fine structures at low momentum which the authors focus on are really relevant or require unrealistically large system sizes to become observable.

The Reviewer is correct about the size of the grating that are 50 μm along y and 300 μm along x (we follow here the convention on the axis given in Figure 1 of the paper, where we have added the dimensions in the revised caption). That being said, the main challenge to the observation of an anisotropic superfluidity in our system is probably the fact that the condensate is in a state at $k = 0$ with zero group velocity and therefore it is not propagating.

In fact, in the peculiar system that we consider (the condensate in the bound-state-in-the-continuum), any kind of flow along one or the other direction cannot be realized at all without getting out of this symmetry protected state: the condensate cannot be excited with a non-zero momentum, as any non-zero k on the saddle-like polariton dispersion will not correspond to the BIC state any longer. To probe anisotropic superfluidity in this case would require moving a defect against the resting BIC condensate. This however is hard to realize (even a laser creating an *ad hoc* potential cannot be easily swiped at such a high frequency) and would require a completely different experiment compared to the one discussed in the current paper. Here, measuring the details of the dispersion of collective excitations and comparing it to the theory, which provides an accurate description for the smallest- k region, is already the proof of concept of direction-dependent sound velocity. Obtaining this insight even theoretically allows one to devise directions for future investigation of anisotropic superfluidity. In particular, we are planning to create the BIC state off the Γ -point of the folded bands, in which case a finite velocity could be given to the superfluid. We did some simulations

showing that for a carefully chosen step of the grating, the crossing of the TE0 mode and the corresponding counter-propagating TE1 happens at a finite k of the order of $1 \mu\text{m}^{-1}$. The realization and the characterization of this new kind of hybrid off- Γ BIC is well beyond the scope of this work (it would require new sample design where the BIC will be parametric, and would only lead to propagation in one direction), but we prefer to mention it here as a future strategy to overcome a part of the experimental challenges mentioned by the Reviewer.

- Along similar lines, asymmetric sound velocities and superfluid properties certainly are very interesting. However, asymmetric superfluid properties have of course been found in other systems as well and it would certainly be interesting to compare the authors' results to these other systems. Perhaps one of the most prominent examples of anisotropic superfluids is given by dipolar quantum gases, e.g., Physical Review Letters 121, 030401 (2018).

Once again we thank the Reviewer for this useful remark. Clearly, anisotropic superfluidity was discussed in other contexts, including dipolar BECs of cold atoms. In this case, however, the observed effects of anisotropic transport are connected to the different strength of the roton softening in different directions. The suggested Reference [new Ref. 33], for example, reports on the observation of anisotropic *critical* velocity as compared to the sound velocity in the system, due to the dipolar polarization implied by the magnetic field. Nevertheless, the anisotropy of the sound velocity in this case is clearly relevant to our discussion. In fact, the anisotropy of sound velocity reported in our work can be better compared with anisotropic superfluidity reported for 2D excitons in a periodic potential and cavity photon BECs in periodically modulated microcavities. We have added the corresponding discussion to the conclusions in page 6, and the new References 31-35 to the revised bibliography.

-The authors' statement that "we believe that this study could open a new way of controlling the condensate properties by engineering its excitation spectrum" is a bit vague. Do they have a concrete plan on how to engineer the excitation spectrum and how to utilize this to control the condensate? Which properties do they intend to control and why?

What we meant by this sentence is that for a bound in the continuum condensate, the excitation spectrum inherits some properties of the grating producing the BIC. As a consequence, the excitation spectrum of a BIC in a 1D grating shows the specific properties demonstrated in the current work (e.g. parallel stripes whose extension is related to the population of the condensate).

We expect that by using a different grating symmetry, for instance a 2D grating, the properties of the excitation spectrum must be different. It is also worth mentioning that the topology of the photonic mode is transferred to the condensate, so grating having different topologies will result in condensates having also different topologies. A short comment on this has been added to the revised Discussion.

In summary, polariton BECs in periodically modulated waveguide geometries featuring BICs are still a relatively young topic and therefore a study of their excitation spectra is certainly relevant to the field - even although the results more or less clearly show that superfluidity will not be an issue here. Still, the saddle point condensation geometry will likely lead to interesting physics in the future. However, as pointed out above, partially the analysis seems a bit shaky and might benefit from a more thorough discuss and a clearer presentation of the data at hand.

In the revised version of the manuscript, we have added some new data and new analysis and, with the help of many valuable comments, improved the presentation of some of the results and provided more thorough comments and discussions. We believe that, thanks to the Reviewers' suggestions, the presentation of both theoretical and experimental results is much more solid now.

The added analysis in fact reveals that not only the PL shape directly follows the theoretically calculated curvature of the Bogoliubov dispersion, but also that the parallel stripes appearing in the far field PL patters correspond to the states lying at the same energy with the condensate, while values of k_x closer to the BIC correspond to above-condensate excitations and are less populated. Clearly, this physics is new and have not been previously reported, to

the best of our knowledge, neither for polariton condensates, nor for any other systems. As the Reviewer points out themselves, the system under study is new and leads to some new unexplored physics.

Concerning the superfluidity discussion, in the revised version of the paper we clearly state the experimental limitations in the current setting, while still provide the theoretical discussion. The main focus of the paper is however now on the observed peculiarities of the Bogoliubov spectrum and the underlying physics.

Also, some of the outlooks used for motivation might give the impression of being slightly overselling the results - also as outlined above. It is my impression that a revised manuscript that improves the explanation and presentation of the results and thereby makes it clearer that the evidence for the postulated claims is sound and also chooses a more moderate and realistic tone in terms of the outlook and the potential further research directions arising from the present work may be a reasonable addition to the literature and also a manuscript that reaches the standards of Nature Communications. However, I have doubts that the current revision of the manuscript meets these standards. Accordingly, my suggestion would be to reconsider a revised manuscript.

We hope to have addressed in our reply above and with the majorly revised version of the paper, that contains now some improved explanations and a clearer presentation along with new data and analysis, the concerns raised by the Reviewers. The claims of our work are now clearer and substantiated by the results presented, and potential further research outlined. Thus we believe that the Reviewer will now be inclined to recommend the revised manuscript for publication.

Some minor points:

- On page 3, discuss Fig. 1c and d where the PL emission below threshold is shown and state that the upper of the two polariton levels shown is more populated than the lower one. Are the authors sure about that? I would expect the upper levels to be extremely leaky, while the lower levels should be rather long-lived, so I am not sure that the magnitude of the PL directly reflects the magnitude of the population. Or does the lifetime already vary drastically along the lower branch?

Since the excitation of the system is non-resonant, the excitons forming from the incoherent reservoir first relax to the exciton dispersion. Unlike in the regular microcavity polariton case, the exciton line in our case lies above the upper of the lower two polariton branches (marked ULP in Supplemental Figure 1) and it can be seen very broadened above the ULP in Figs. 1c-f. From there the particles scatter towards the ULP and LLP. The images in Fig. 1c and d are time-integrated and taken below threshold, which means that the lowest branch is not yet macroscopically occupied compared to the upper one. Since the lifetime on the LLP is much longer than on the ULP, the ULP shines brighter (because of the fast recombination of particles), see the left-hand side of Fig. 1e. At the same time the particles keep relaxing towards the LLP. At higher powers, as soon as the threshold is reached, the population on the ULP is much lower compared to that of the condensate in the saddle point of the LLP (see the right-hand side of Fig. 1e). At the same time, as soon as one steps aside from the near-condensate bright spot, one can again compare the populations of the lower and upper branches (the right-hand side of Fig. 1f; still, the LLP branch is more occupied here due to the excitations from the condensate than the ULP).

- On page 3 of the supplementary materials some parts seem to be missing. In the right column, the sentence starting directly beyond the only equation in the column starts with "equations one sees[...]" which seems to indicate that the sentence is incomplete.

We thank the Reviewer for noticing this error, which has been corrected in the revised version of the Supplementary Notes, along with some other misprints.

REVIEWERS' COMMENTS

Reviewer #1 (Remarks to the Author):

I appreciate the efforts of the authors in addressing my comments. In particular, I think the new data and the new discussion of the results better highlights the new physics explored by this work. On one hand, this manuscript describes new physics of BEC in an emerging system of exciton-polaritons. On the other hand, I am not sure this topic is general enough to meet the expectation of the the broad readership of Nature Communications. I still find the manuscript very technical. Nevertheless, I think that the quality of the research deserves publication. If the editors believe that the topic is suitable, I support publication in Nature communications.

Reviewer #2 (Remarks to the Author):

The authors managed to address all the critics raised by me and, as I far as I can judge, from other referees. They did great job preparing their manuscript, amending it, and answering the questions. The manuscript, indeed, deserves to be published in nature Comm. Thus, I strongly support its publication.

Reviewer #3 (Remarks to the Author):

The authors have resubmitted a revised version of their manuscript on the excitation spectrum of a dark polariton condensate based on a bound state in the continuum.

In the first round of reviews, the referees' opinions were diverse. and included the opinions that the manuscript is of high quality, but feels incremental and should focus more on the unique properties of BIC condensates (referee 1), that the results look solid, but do not result in broad interest (referee 2) and that the manuscript is in principle suitable for Nature Communications, but a revision of the manuscript is required to substantiate the authors' claims (referee 3).

In response to these reviews, the authors performed two major changes: First, they added new cw measurements which allow the reader to identify the shape of the observed dispersion even more clearly and thus result in a better comparison between theory and experiment. Second, they slightly moved the focus away from the anisotropic sound velocity towards the shape of the excitation spectrum. They used the results of the cw experiments to provide further insights into the asymmetric excitation spectrum and now also explain why the PL maximum occurs at a certain k_x in terms of the

parts of the spectrum which are located at the same energy as the ground state, which is an interesting and nontrivial finding.

Therefore, the authors have certainly provided a satisfying response to my (referee 3) comments. In my opinion, the results presented here are very interesting from at least two points of view: the fact that the condensate itself is optically dark opens up the possibility to study the excitation spectra of polariton condensates even at very small momenta which are usually hard to investigate due to the brightness of the condensate. Second, the inherent asymmetry of the excitation spectrum will render it possible to tailor the system geometry which will open up the possibility to perform many further experiments such as the parametric BIC hinted at by the authors in the response to the referees.

In my opinion the response to referee 1 is also satisfying. Referee 1 already considered the results as sound and asked for a more detailed description of physics unique to BIC condensates, which is what the authors provided.

I also consider the authors' response to the technical comments of referee 2 satisfying. I cannot comment on the non-technical aspects of the authors' response to referee 2 as referee 2 only stated that it is difficult for him to expect a broad interest of this manuscript without providing any further detailed reasoning or explanation of this statement which renders it impossible to respond to this statement properly.

In summary, it is my opinion that the manuscript has improved significantly. When considering the question of broad interest, I consider the novelty presented and the number of people that may become interested in the system at hand as the correct indicator rather than the number of people already working on it - the latter is a better indicator for hype. The system investigated here is a relatively novel one which has not been investigated in much detail so far, but offers a lot of interesting experimental possibilities by tailoring the system, so I indeed expect these results to be of broad interest. Therefore, I support the publication of this manuscript in Nature Communications.

A minor point:

- There seems to be a typo in the caption of figure 3, there is no caption for panel (e), but panel (c) is discussed again in the end of the caption. I assume that the authors intended to refer to (e) here.

Authors' response to the Reviewers' comments

We would like to thank all the Reviewers for carefully considering our revised manuscript and our reply to their previous concerns and suggestions, certainly they all helped to improve the quality of the paper. We thank the Reviewers for supporting its publication in Nature Communications.

Below we provide the answer to the only point raised by the Reviewer #3:

A minor point:

- There seems to be a typo in the caption of figure 3, there is no caption for panel (e), but panel (c) is discussed again in the end of the caption. I assume that the authors intended to refer to (e) here.

We thank the Reviewer to pointing out this misprint that we have corrected in the final revision of the manuscript.